# Organo-Vermiculites Modified by Aza-Containing Gemini Surfactants: Efficient Uptake of 2-Naphthol and Bromophenol Blue

**DOI:** 10.3390/nano12203636

**Published:** 2022-10-17

**Authors:** Jianchao Gong, Tingting Wang, Wei Zhang, Lin Han, Mingxiao Gao, Tianen Chen, Tao Shen, Yaxiong Ji

**Affiliations:** 1Hoffmann Institute of Advanced Materials, Shenzhen Polytechnic, 7098 Liuxian Boulevard, Nanshan District, Shenzhen 518055, China; 2Innovation Laboratory of Materials for Energy and Environment Technologies, Department of Physics, College of Science, Tibet University, Lhasa 850000, China; 3PetroChina Petrochemical Research Institute, Beijing 102206, China; 4Anshan No. 1 Middle School, Anshan 114051, China

**Keywords:** organo-vermiculite, aza-containing gemini surfactant, 2-Naphthol, bromophenol blue, adsorption

## Abstract

To explore the effect of spacer structure on the adsorption capability of organo-vermiculites (organo-Vts), a series of aza-containing gemini surfactants (5N, 7N and 8N) are applied to modify Na-vermiculite (Na-Vt). Large interlayer spacing, strong binding strength and high modifier availability are observed in organo-Vts, which endow them with superiority for the adsorption of 2-naphthol (2-NP) and bromophenol blue (BPB). The maximum adsorption capacities of 5N-Vt, 7N-Vt and 8N-Vt toward 2-NP/BPB are 142.08/364.49, 156.61/372.65 and 146.50/287.90 mg/g, respectively, with the adsorption processes well fit by the PSO model and Freundlich isotherm. The quicker adsorption equilibrium of 2-NP than BPB is due to the easier diffusion of smaller 2-NP molecules into the interlayer space of organo-Vts. Moreover, stable regeneration of 7N-Vt is verified, with feasibility in the binary-component system that is demonstrated. A combination of theoretical simulation and characterization is conducted to reveal the adsorption mechanism; the adsorption processes are mainly through partition processes, electrostatic interaction and functional interactions, in which the spacer structure affects the interlayer environment and adsorptive site distribution, whereas the adsorbate structure plays a role in the diffusion process and secondary intermolecular interactions. The results of this study demonstrate the versatile applicability of aza-based organo-Vts targeted at the removal of phenols and dyes as well as provide theoretical guidance for the structural optimization and mechanistic exploration of organo-Vt adsorbents.

## 1. Introduction

Gemini surfactants (Geminis) are amphiphilic molecules with two hydrophobic alkyl chains linked by a specific spacer group, which own more unique physical–chemical properties than the monomeric one, such as surface activity, antimicrobial property, self-assembling ability, etc. [1]. Benefitting from the low toxicity, facile synthesis and adjustable hydrophobicity, cationic Geminis are widely used in corrosion inhibition, enhanced oil recovery, material synthesis and pollutant adsorption [2]. In the field of wastewater treatment, hybrid organic/inorganic materials can be fabricated by immobilizing cationic Geminis on specific negatived charged precursor, modulating the material structure and property to meet specific actual wastewater treatment [3,4].

Natural clays (such as montmorillonite, bentonite and vermiculite) are 2:1 layered phyllosilicates; they are shaped by layered structure and characterized by negatively charged surfaces [5]. Among them, vermiculite (Vt) is a magnesium-containing hydroaluminosilicate secondary metamorphic mineral [6]. Due to its abundant reserve, low cost, and superior properties over other clay minerals (higher layer charge and better adsorption capacity), Vt is a promising inorganic precursor for preparation of organic adsorbents [7]. Since most organic pollutants are hydrophobic, the hydrophilic nature of raw Vt limits its adsorption capacity. Decoration of cationic Geminis on Vt, therefore, helps to form a hydrophobic surface and expand the interlayer space, contributing to increased affinity of the resultant organo-vermiculites (organo-Vts) toward organic pollutants [8,9]. Organo-Vt based adsorbents have been considered for their economic viability and sustainability in several aspects. (i) In terms of the raw material, Vt is a typical low-cost and abundant clay mineral, and the modifier gemini surfactants are low-cost. Both the inorganic and organic raw materials are cost-effective and environmental-friendly sources characterized by “resource cost”. (ii) In terms of the preparation process, organic modification is realized by a simple wet process without high temperature nor high pressure. After organic modification, almost all the added modifiers are effectively loaded and utilized [10]. Therefore, the preparation processes follow the principle of “energy cost”. (iii) In terms of the adsorption process, organo-Vts own the comprehensive capabilities of excellent adsorption, stable regeneration and easy separation [11]. These properties not only support a sustainable adsorption cycle, but also effectively avoid the generation of hazardous by-products. The adsorption processes of organo-Vts in this study indicate the concept of “environmental cost”.

Benefitting from the structural properties of both the precursor and modifier, the adsorption mechanisms of organo-Vts are mainly through intraparticle diffusion (derived from the natural structure of the precursor), the partition process (derived from the alkyl chain of the modifier) and the functional interactions between the modifier and target pollutant (such as the π–π/OH–π/NH–π interaction derived from the functional group of the modifier) [12]. During the adsorption processes of organic pollutants on organo-Vts, the structure and functional group in the modifier play important roles for the improvement of adsorption capacity of organo-Vts. Yu et al. explored the adsorptivity of organo-Vts with unsaturated functional groups and found that the superiority of common surfactants for adsorption enhancement was due to the suitable interlayer environment and strong hydrophobicity of the resultant organo-Vts [13]. Compared with rigid spacers, therefore, cationic Geminis with flexible functional groups are more suitable. Wang et al. introduced imino-containing cationic Geminis into organo-Vts and manifested the existence of NH–π interaction between organo-Vts and RhB, with a strength that is stronger than that of π–π and OH–π interactions [14]. Despite that the well-developed organo-Vt adsorbents, aza- and imino-substituted gemini surfactants are expected to be suitable modifiers for novel organo-Vts, there is hardly any research about the application of aza-substituted gemini surfactants as modifiers onto Vt.

Persistent organic pollutants (such as phenols, pesticides, etc.) and dyes are usually degradation-resistant, causing serious damage to living organisms and modern society [1]. Bromophenol blue (BPB) and 2-naphthol (2-NP), as typical organic pollutants with relatively low solubility in water, have been subjected to photo degradation with poor recyclability [15]. Even trace existence of BPB or 2-NP in aquatic environment would cause nonnegligible teratogenic and carcinogenic damages [16]. Real wastewater usually contains multi-component pollutants, which increases the treatment difficulty due to various pollutant structures and properties [17]. Organo-Vts have a proven feasible adsorption capacity and recyclability toward both dyes and phenols, the applicability of which, in a multi-component system, results from secondary intermolecular interactions between pollutant molecules (such as π–π stacking between the adsorbates adsorbed and dissolved in solution) [12]. Therefore, insights into the adsorptivity–structure relationship and the adsorption mechanism of aza- and imino-containing organo-Vts deserve further investigation 

In order to enrich the existing organo-Vt adsorbent species and elucidate the effect of the spacer intervals within modifier on the adsorption property of organo-Vts, novel derivatives of gemini surfactants with aza-containing spacers (12-5N-12, 12-7N-12 and 12-8N-12) are imported as a modifier onto Na-Vt for the first time. The thermal stability, morphology and packing density of the synthesized organo-Vts are examined using a series of characterizing techniques. The adsorption kinetics, isotherms, thermodynamics and structure–adsorptivity relationship are calculated through the discussion of batch adsorption experiments, and the adsorption mechanisms are explored by the theoretical simulation and characterization of spent samples. The regenerability of organo-Vts is explored through the simple acid pickling method.

## 2. Materials and Methods

### 2.1. Materials

Vermiculite (Vt) was bought from Shanghai Aladdin Biochemical Technology Co., Ltd. (Shanghai, China) and treated by Na_2_CO_3_ (Shanghai Aladdin Biochemical Technology Co., Ltd., Shanghai, China). CEC is the cation exchange capacity of vermiculite (Vt), which means the amount of exchangeable cation within the interlayer/on the surface of Vt. The CEC of vermiculite is 126 mmol/100g, representing that the exchangeable cations are 126 mmol per 100 g of Vt [10]. Source materials for modifier synthesis, N,N,N′,N′,N″-pentamethyl diethylenetriamine (PMDA), 3,3′-iminobis(N, N-dimethyl propyl-amine) (IDMPA), 1,1,4,7,10,10-hexamethyl-triethylenetetramine (HMTA) and 1-bromododecane (BD-12) were supplied by Aladdin Inc., Shanghai, CN The 2-Naphthol (2-NP) and bromophenol blue (BPB) were also bought from Aladdin Inc., Shanghai, CN. Acetonitrile and ethyl acetate were purchased from Macklin. All chemicals were used without further purification.

### 2.2. Synthesis of Gemini Surfactants and Organo-Vts

Series of aza-containing ß (1,5-bis(dodecyl)-1,1,3,5,5-pentamethyl-3-aza-1,5-pentanediammonium dibromide, 5N), (1,9-bis(dodecyl)-1,1,5,9,9-pentamethyl-5-aza-1,9-nonanediammonium dibromide, 7N) and (1,10-bis(dodecyl)-1,1,4,7,10,10-hexamethyl-4,7-diaza-1,10-dodecanediammonium dibromide, 8N) were synthesized according to the previous literature [18]. Briefly speaking, 2.2 mol of BD-12 was dissolved in 100 mL of acetonitrile under continuous stirring. Slowly, 1 mol of PMDA (IDMPA or HMTA) was added into the mixture and reacted at 85 °C for 24 h under refluxing. After removal of the solvent and recrystallization with ethyl acetate several times, the final products 5N, 7N and 8N were obtained, and their chemical structures are shown in Figure 1a.

The modification of Na-Vt by surfactants was conducted by mixing 1 g of Na-Vt and certain amounts of surfactants (0.252, 0.504, 0.756, 1.008 and 1.26 mmol) in 50 mL of deionized water, and the mixtures reacted under 60 °C for 4 h. The resultant organo-Vts (5N-Vt, 7N-Vt and 8N-Vt) were characterized by FT-IR, XRD, TG-DTG, SEM and EA, as well as detailed operation procedures, as described in the Appendix A.

### 2.3. Adsorption Experiment

Batch adsorption experiments were adopted for evaluating the adsorption performance of organo-Vts. Analysis about the influence factor (modifier dosage, adsorption time, initial CR concentration, temperature and solution pH) and adsorption mechanism (adsorption kinetics, isotherms and thermodynamics) was conducted by mixing 0.03 g of organo-Vts and 30 mL of adsorbate solution under varying conditions, respectively. The concentration 2-NP and BPB is measured by UV–vis. The UV–vis adsorption bands of bromophenol blue (BPB) and 2-naphthol (2-NP) are 592 and 274 nm, respectively. Due to the limited measure range of UV–vis, the adsorbance of standard BPB and 2-NP solution (0.1 mmol/L) are 2.353 and 0.424, respectively. The adsorption amounts (*q*_e_, mg/g) were obtained by the following equation:(1)$$q_{e}=\frac{C_{0}-C_{e}}{m}V$$
where *q*_e_ is the adsorption amount onto adsorbent (mg/g), *C*_0_ and *C*_e_ are the initial and equilibrium dye concentrations (mg/L), respectively, m represents the mass of adsorbent (g) and *V* stands for the volume of solution (L).

### 2.4. Adsorption Kinetics, Isotherms and Thermodynamics

The equations of pseudo-first order (2), pseudo-second order (3) and intra particle diffusion (4) models; the Langmuir (5), Freundlich (6) and Redlich-Peterson (7) isotherms; and thermodynamic parameters (8 and 9) are expressed as follows:(2)$$\log\left(q_{e}-q_{t}\right)=\log q_{e}-\frac{k_{1}}{2.303}t$$
(3)$$\frac{t}{q_{t}}=\frac{1}{k_{2}q_{e}{}^{2}}+\frac{1}{q_{e}}t$$
*q*_t_ = *k*_i_*t*^1/2^ + *C*
(4)where *k*_1_ (min^−1^) and *k*_2_ (g (mg min)^−1^) were the pseudo-first-order and pseudo-second-order rate constants, respectively. *q*_e_ (mg g^−1^) and *q*_t_ (mg g^−1^) were the adsorption capacities at equilibrium and at time *t* (min), which represents contact time, respectively. All these unknown parameters can be determined from plots of log (*q*_e_ − *q*_t_) against t and *t*/*q*_t_ against *t*. *k*_id_ (mg g^−1^ min^−1/2^) is the rate constant of the intraparticle diffusion kinetic model, and the values of *C* and kid can be determined from the intercept and slope of the linear plotted of *q*_t_ against *t*^1/2^, respectively.

The Langmuir, Freundlich and Redlich-Peterson isotherms were expressed as follows:(5)$$q_{e}=\frac{Q_{m}K_{L}C_{e}}{1+K_{L}C_{e}}$$
(6)$$q_{e}=K_{f}C_{e}^{1/n}$$
(7)$$q_{e}=\frac{AC_{e}}{1+BC_{e}^{g}}$$
where *q*_e_ (mg/g) is the adsorption capacity onto per unit mass of adsorbent at equilibrium, *C*_e_ is the solute equilibrium concentration (mg/L), and *q*_m_ (mg/g) is maximum adsorbed amount in the theoretical. *k*_L_ (L/mg), *k*_f_ (mg/g) and n represent the constants of Langmuir and Freundlich, respectively. A (L/g) and B ((L/mg)g) are the Redlich–Peterson model constant. g fluctuated between 0 and 1 with two limiting behaviors: Langmuir form for *g* = 1 and Henry’s law form for *g* = 0.

Thermodynamic parameters could be calculated using the following equations:(8)$$\ln K_{L}=\frac{\Delta S{}^{\circ}}{R}-\frac{\Delta H{}^{\circ}}{RT}$$
(9)$$\Delta G{}^{\circ}=-RT\ln K_{L}$$
where *K*_L_ is the Langmuir constant, *q*_e_ and *C*_e_ have the same definitions with above equations. *R* is the universal gas constant (8.3145 J/(mol K)) and T represents the absolute temperature in Kelvin. The values of Δ*H*° and Δ*S*° can be extrapolated from intercept and slope of ln *K*_L_ versus 1/T.

### 2.5. Theoretical Calculation

Quantum chemical calculations were conducted on Material Studio 8.0. The optimized structures and molecular frontier orbitals were calculated based on density functional theory (DFT). Dmol3 module was adopted for calculation, where the function GGA&BLYP (generalized gradient approximation & Becke–Lee–Yang–Parr) was used in all processes. The Adsorption Locator module was applied to simulate the optimized adsorption configuration, and the forcefield was set to universal. The molecular structures of pollutants and adsorbents are plotted in MS based on the formulas and then optimized using the DMol3 module. The adsorption structures are calculated based on the calculated molecules using the adsorption locator module. According to the DFT method, the adsorption behavior between two molecules is calculated, so the calculated size is the size of two molecules. The alkyl chain length of 5N, 7N and 8N is approximately 1.54 nm, and their spacer lengths are 0.42, 0.44 and 0.51 nm, respectively. The width and height of BPB/2-NP are about 1.07/0.71 and 0.92/0.50 nm, respectively.

## 3. Results and Discussion

### 3.1. Characterization

#### 3.1.1. FT-IR Analysis of Surfactants and Organo-Vts

The FT-IR spectra of 5N, 7N and 8N are presented in Figure 1b. Peaks at 3426 cm^−1^ are related to the vibrations of physically adsorbed water molecules, with bending vibrations that are observed at 1630 cm^−1^. The stretching vibrations of -C-H (derived from -CH_2_ and -CH_3_ in the modifier alkyl chain) are observed at 2922 and 2852 cm^−1^, and the bending vibration of -C-H occurs at 1468 cm^−1^, which overlaps with the vibration of the aza group [19]. To further clarify the surfactant structure, the data of EA are listed in Table 1, and the data of EA and ^1^H NMR are listed in Appendix A. The results of FT-IR, EA and ^1^H NMR demonstrate that the synthesized 5N, 7N and 8N are target products with a high purity.

The FT-IR spectra of Na-Vt and organo-Vts are shown in Appendix A and Figure 1c. The hydrophilic nature of Na-Vt is verified by the strong peak at 3444 and 1635 cm^−1^ (derived from physically adsorbed water molecules). Peak at 993 cm^−1^ is caused by Si-O vibration. In the FT-IR spectra of organo-Vts, additional peaks at 2920/2849 and 1465 cm^−1^ are characteristic of 5N, 7N and 8N, indicating the existence of modifiers in organo-Vts [20]. Compared with the vibration of -CH_2_ in Figure 1b (2922/2852 and 1468 cm^−1^), variation of these peaks to lower wavenumbers results from the confinement effect of the Na-Vt interlayer on the alkyl chain configuration [20]. FT-IR results indicate that 5N, 7N and 8N are successfully intercalated into or adsorbed on the interlayer space of Na-Vt, which can increase the hydrophobicity of organo-Vts.

#### 3.1.2. XRD Patterns of Organo-Vts

The XRD patterns of organo-Vts are shown in Figure 2a–c. Na-Vt owns the *d*_001_ spacing of 1.12 nm, providing adsorptive sites for accommodating water molecules and exchangeable cationic ions (Appendix A) [21]. After organic modification, organo-Vts show sharply increased interlayer spacings due to the intercalation of surfactants. Under a low modifier dosage (0.2 CEC), the interlayer spacings of 5N-Vt, 7N-Vt and 8N-Vt are 3.67 (2.77 and 1.19), 3.56 (1.20) and 3.67 (1.21) nm, respectively, indicating the effective expanding effect of modifier on the interlayer environment. The corresponding interlayer distances of 5N-Vt, 7N-Vt and 8N-Vt, calculated by subtracting the thickness of Na-Vt from the interlayer spacings, are 2.71 (1.81 and 0.23), 2.60 (0.24) and 2.71 (0.25) nm, respectively. Increased interlayer distances are observed with the modifier dosage increasing, with the largest values being 2.84 (1.86 and 0.29), 3.11 (0.4) and 3.00 (2.00 and 0.28) nm for 5N-Vt, 7N-Vt and 8N-Vt (the modifier dosage is 1.0 CEC), respectively.

Possible modifier arrangements within the interlayer space of 5N-Vt, 7N-Vt and 8N-Vt could be deduced from the combined analysis of modifier size and the interlayer distance of organo-Vts [21]. The alkyl chain length of 5N, 7N and 8N is approximately 1.54 nm, and their spacer lengths are 0.42, 0.44 and 0.51 nm, respectively. Different binding manners and modifier arrangements contribute to various interlayer distances in organo-Vts (Figure 2d): (i) The distances of 2.71 and 2.60 nm correspond to the paraffin bilayer arrangement, with both of the modifier headgroups binding on the same side of the Na-Vt plane. Increased title angles between the modifier alkyl chain and Na-Vt plane result in the enlarged interlayer distance under higher modifier dosage. (ii) The distances of 1.81, 1.86 and 2.00 nm correspond to the paraffin bilayer arrangement, with one of the modifier headgroups binding on the surface of Na-Vt, while the other headgroup is connected to the alkyl chain directing to the interlayer of organo-Vts. (iii) The interlayer distances of 0.23, 0.29, 0.24, 0.28 and 0.25 nm correspond to the lateral bilayer arrangement of modifiers. Compared with the configurations of 5N and 8N, the alkyl chains of 7N prefer to be bonded on the surface of Na-Vt with smaller title angles, indicating the great affinity of 7N to Na-Vt. Examining the variation of d001 spacing of organo-Vts, 0.4 CEC is chosen as the optimal modifier dosage for further characterization and adsorption tests.

#### 3.1.3. TG-DTG Curves and EA Analysis of Organo-Vts

The TG-DTG curves of organo-Vts are shown in Figure 3a,b. As shown in Figure 3a, the weight losses of organo-Vts below 100 °C are due to the loss of physically adsorbed water molecules, with amounts that are lesser than those of Na-Vt (Appendix A), indicating the increased hydrophobicity of organo-Vts due to organic modification [22]. Compared with the TG curve of Na-Vt, additional weight losses of organo-Vts between 200–400 °C are caused by the decomposition of modifiers, which verifies the successful organic modification process.

The interactions between surfactants and organo-Vts contain physical adsorption of surfactants within the interlayer space of Na-Vt, electrostatic interaction between the surfactant and Vt and the van der Waals force between surfactant molecules, which corresponds to different weight loss temperatures [23]. In the DTG curve of organo-Vts (Figure 3b), the decomposition temperatures of 5N-Vt, 7N-Vt and 8N-Vt are 211.1 (359.0), 222.6 (349.6) and 187.0 °C, respectively. The first weight loss stage verifies the desorption of physically adsorbed surfactants, where 7N-Vt shows the highest desorption temperature, which may be due to the strongest intermolecular interactions between the adsorbed 7N molecules. The second weight loss stage corresponds to the decomposition of electrostatically intercalated modifiers in Na-Vt; the strength of electrostatic interaction is affected by the steric hindrance of the head group of surfactant molecules (demonstrated by the highest decomposition temperature of 5N-Vt). Moreover, the amounts of electrostatically bonded 7N molecules are the largest, as verified by the largest weight losses of 7N-Vt at around 350 °C.

Combining the results of TG-DTG and EA, the following conclusions can be obtained: (i) The total C, H and N contents in 5N-Vt, 7N-Vt and 8N-Vt, as calculated from EA analysis, are 17.06 wt%, 16.80 wt% and 16.32 wt%, respectively, which are in line with the total weight losses in the TG curves (approximately 19.0 wt%, 17.3 wt% and 16.6 wt% corresponding to 5N-Vt, 7N-Vt and 8N-Vt, respectively). (ii) The modifier loadings calculated by the N content of 5N, 7N and 8N are 0.502, 0.501 and 0.435 mmol on 1 g of Na-Vt, respectively. (iii) Modifier availability represents the utilization ratio of the modifier during the modification process, which can be calculated by the ratio of the modifier loaded (calculated by TG-DTG and EA of the organic contents in organo-Vts) to the modifier added (determined by the CEC of Vt). The higher modifier availability is in line with the greater modifier utilization and lesser modifier loss in the organic modification process. The values of modifier availability (the mole ratio of modifier loaded to added) for 5N-Vt, 7N-Vt and 8N-Vt are 99.60%, 99.40% and 86.31%, respectively.

#### 3.1.4. SEM Images of Organo-Vts

The morphology of Na-Vt is presented as a typical flat surface with small fragments stacking irregularly (Appendix A). After organic modification, the surface of organo-Vts becomes fluffy and curled (Figure 3c–e), indicating the coverage effect of surfactants on the surface of Na-Vt. Both 5N-Vt and 8N-Vt retain the lamellar structure and stacking layers of the raw Na-Vt, which may be due to the modifiers being mainly adsorbed on the surface of Na-Vt, without destroying the lamellar structures. In the case of 5N-Vt, on the other hand, the layers stack irregularly and loosely, proving the more preferable adsorption trend of 7N in the interlayer space of Na-Vt.

### 3.2. Adsorption Test

#### 3.2.1. Effect of Modifier Dosage on Adsorption

The weak affinity of Na-Vt to 2-NP and BPB is indicated by their negligible adsorption amounts (15.86 and 30.41 mg/g, respectively) for Na-Vt. After organic modification, organo-Vts exhibit enhanced the adsorption capability toward both 2-NP and BPB, even at a low modifier dosage, with the adsorption capacities of 5N-Vt, 7N-Vt and 8N-Vt being 76.62, 117.37 and 115.83 mg/g toward 2-NP or 84.25, 85.45 and 93.13 mg/g toward BPB under the modifier dosage of 0.2 CEC, respectively.

As shown in Figure 4, the adsorption capacities of organo-Vts increase with the modifier dosage increasing from 0.2 CEC to 0.4 CEC and follow by a gradual equilibrium or slight decrement at a modifier dosage higher than 0.6 CEC. Notably, the maximum adsorption capacities of organo-Vts are obtained at 0.4 CEC, and the adsorption amounts of 2-NP/BPB for 5N-Vt, 7N-Vt and 8N-Vt are 120.31/121.52, 126.16/123.46 and 116.77/120.39 mg/g, respectively. In the case of 2-NP adsorption, similar adsorption trends indicate the similar uptake mechanisms on organo-Vts, which may be the hydrophobic interaction derived mainly from the modifier alkyl chains. In the BPB adsorption process, however, the decreased trend of BPB uptake for 8N-Vt may be caused by the greater effect of steric hindrance on the diffusion of BPB into the interlayer of 8N-Vt. Notably, 7N-Vt shows the highest adsorption capability toward both 2-NP and BPB, which is in line with its largest interlayer spacings in the XRD patterns.

Combining the aspects of adsorption capacity and the mole ratio of the adsorbate to the added surfactant molecules, the modifier dosage of 0.4 CEC is chosen as the optimal amount and used in the following adsorption experiments.

#### 3.2.2. Effect of Time and Adsorption Kinetics

The effects of the time and adsorption kinetics of 2-NP and BPB are shown in Figure 5. Organo-Vts show a rapid adsorption rate toward 2-NP, with the equilibrium time reaching approximately 30 min. In the adsorption process of BPB, a slow adsorption equilibrium is obtained within 60 min. To clarify the adsorption kinetics, pseudo-first order (PFO), pseudo-second-order (PSO) and intraparticle diffusion (IPD) models are applied to fit the adsorption data (Table 2) [24]. Compared with PFO, PSO models are more suitable to the equilibrium data, with a higher *R*^2^ and better consistency of *q*_e,cal_ to *q*_e,exp_, indicating that the adsorption processes are mainly chemically controlled. The smaller values of *k*_2_*q*_e_ (second-order rate index, indicating the inverse of the half-life of adsorption process) are in line with the faster adsorption rate [25]. Moreover, 8N-Vt owns the highest adsorption rate toward both 2-NP and BPB, indicating the greater affinity of the aza-group to organic pollutants than the alkyl chain. Notably, the uptake rate of BPB for 5N-Vt is higher than 7N-Vt, which may be due to the depressed effect of the steric hindrance of BPB on the adsorption rate.

The participation of the diffusion process is verified by the obvious division of the IPD model into three parts (corresponding to the external diffusion, intraparticle diffusion and equilibrium stages) [26]. The following conclusions can be obtained from the IPD model: (i) The high values of *R*^2^ in the second stage (Table 2) indicate the participant role of intraparticle diffusion during 2-NP/BPB adsorption. (ii) In the IPD model of 2-NP adsorption, the highest values of *k*_1_ and the lowest values of *C*_1_ indicate that surface adsorption is the control step in the whole 2-NP adsorption process. (iii) The important role of the diffusion process for BPB adsorption is indicated by the highest *k*_2_ and the lowest *C*_2_ values. The agreeable diffusion rate and *k*_2_*q*_e_ value of BPB adsorption on 8N-Vt between the IPD and PSO models proves that intraparticle diffusion may be the main mechanism for BPB adsorption, demonstrating the important effect of suitably packed modifiers within the interlayer space of organo-Vts on BPB adsorption. On the contrary, the inverse rate trend between IPD and PSO indicates the lesser role of the diffusion process in 2-NP adsorption, which may be controlled by specific interactions instead of the diffusion process. From the intraparticle diffusion model of 2-NP and BPB adsorption (Figure 5 c, f), it can be seen that the uptake of 2-NP for organo-Vts is mainly controlled by external adsorption, while external adsorption and the diffusion process both affect the removal of BPB. The slower adsorption process for BPB than for 2-NP is due to the multiple control steps, which results from the differences in the active site and molecule size between 2-NP and BPB.

#### 3.2.3. Effect of Concentration and Adsorption Isotherms

The effects of adsorbate concentration on the adsorptivity are shown in Figure 6. Continuously increased adsorption amounts are observed with the adsorbate concentration increasing. The maximum 2-NP/BPB adsorption under experimental conditions are 142.08/364.49, 156.61/372.65 and 146.50/287.90 mg/g for 5N-Vt, 7N-Vt and 8N-Vt, respectively (*T* = 25 °C). Due to the different molecular weights (M.W.) of 2-NP and BPB, comparing the adsorptivity of organo-Vts by the unit of mg/g may be inaccurate. In terms of the unit of mmol/g for judgment of the adsorptivity, the adsorption capacities of 5N-Vt, 7N-Vt and 8N-Vt toward 2-NP/BPB are 0.99/0.54, 1.09/0.56 and 1.02/0.43 mmol/g, respectively.

Compared with the Langmuir and Redlich Peterson (R-P) isotherms (Table 3 and Table 4), the Freundlich model is more suitable, with all adsorption processes having higher *R*^2^ values, indicating that the adsorbate molecules are prone to bind on the adsorptive sites and form multilayer arrangements [27]. In the Freundlich model, the sequence for the values of *K*_f_ is 7N-Vt > 5N-Vt > 8N-Vt, under the same temperature, indicating the strongest adsorption performance of 7N-Vt toward 2-NP and BPB. In the R-P model, the closer values of *g* to 1 verifies the multilayer arrangement of 2-NP during the whole adsorption process [28]. The adsorbed BPB molecules prefer to be arranged as s monolayer form in the interlayer of organo-Vts, as indicated by the close values of *g* to 0. Different adsorbate configurations may be due to the different molecular sizes of 2-NP and BPB. 2-NP, with a smaller molecular size being more prone to occur with π–π stacking between the adsorbed molecules and those dissolved in solution, thus submitting to a multilayer arrangement during the whole adsorption process. In the case of BPB adsorption, however, the large molecular size limits the secondary interaction (interactions between BPB adsorbed on organo-Vts and those dissolved in solution) between BPB molecules.

#### 3.2.4. Effect of Temperature and Adsorption Thermodynamics

Relatively stable adsorption capacities are observed for organo-Vts with increasing temperature. The related thermodynamic parameters are calculated and shown in Table 5; positive Δ*G*^0^ values demonstrate that all adsorption processes are non-spontaneous [29]. Negative Δ*S*^0^ values indicate the decreased randomness of the adsorption system. The absolute values of Δ*H*^0^ are close to 0, which are in line with the thermodynamic stable nature of all the adsorption processes [30]. From the energetic view, aza-containing organo-Vts own great thermodynamic stability, endowing organo-Vts with great applicability under various adsorption temperatures.

#### 3.2.5. Effect of Solution pH

In order to further investigate the practical applicability of organo-Vts and to explain the adsorption mechanism, the adsorption behavior of 2-NP/BPB under different pH values has been provided in Figure 7. The tested pH values are set as 3, 5, 7, 9 and 11. The zeta potential values of the organo-Vts are measured and listed in Appendix A. As shown in Appendix A, the surfaces of 5N-Vt, 7N-Vt and 8N-Vt remain negatively charged under all tested pH values, with the negative charge increasing under higher pH values.

The adsorption uptake of 2-NP shows a gradual reduction with an increase in pH from 3 to 7. After a slight increment at pH of 9, a sharp decrease in 2-NP uptake is observed under alkaline conditions. BPB retention displays a sharp decrement with the solution pH increasing from 3 to 5, followed by a gradual equilibrium under higher pH values. The p*K*_a_ values of 2-NP/BPB are 9.5/3.0 (4.6), indicating that 2-NP/BPB molecules would be deprotonated and mainly exist as an anionic form when the pH is greater than 9.51/4.6 [31,32]. In case of 2-NP adsorption, a sharp decrease in the adsorption capacity is observed when the solution pH is higher than 9.5, indicating that the existence of electrostatic repulsion between 2-NP and the negatively charged adsorbent surface repress the adsorption process, which is similar in the case of BPB adsorption. Interestingly, a slight increment in the adsorption capacity is observed around a pH value of 8. It can be seen that the negative charges in organo-Vts are slightly decreased at a solution pH around 9, contributing to repressed electrostatic repulsion at a solution pH of around 8. The stable adsorbent surface charge indicates the chemical stability of organo-Vts in the pH range of 5–9. Moreover, from the degree of pH influence on the whole adsorption amounts of 2-NP and BPB, it can be concluded that electrostatic interactions play a vital role in their adsorption processes.

### 3.3. Regeneration

The reusability of 7N-Vt toward 2-NP and BPB is tested by acid pickling. Notably, the adsorption process of 7N-Vt can be stably recycled, even after three cycles, with the adsorption capacities toward 2-NP being 126.16, 120.54, 113.07 and 102.35 mg/g and toward BPB being 123.46, 115.04, 95.32 and 80.21 mg/g, respectively. The 7N-Vt can be regenerated by HCl solution (0.1 mol/L), with a slightly decreased capacity, indicating the emergence of new adsorptive sites, even if some of them are partially lost. Desorption tests demonstrate a sustainable adsorption/desorption cycle for organo-Vts, especially targeted at 2-NP (with the most stable reusability).

### 3.4. Adsorption Feasibility of Organo-Vts in Binary-Component System

The adsorption behavior of 7N-Vt in the binary-component system has been studied, of which the concentrations of 2-NP and BPB are 200 mg/L. After adsorption, the adsorption amounts of 2-NP/BPB for 7N-Vt are 102.28/90.36 mg/g. Compared with the adsorption capacity of 7N-Vt (126.16 and 123.46 mg/g, respectively) in the single-component system, the reduction in the adsorption capacities indicates competitive adsorption with the coexistence of 2-NP and BPB in the solution. The adsorption of 2-NP shows a moderate decrease, while that of BPB exhibits a drastic decrement, which may be due to the easy diffusion of the smaller 2-NP molecules into the interlayer space of 7N-Vt. Moreover, the total number of adsorbed molecules on 7N-Vt in the binary-component system are 0.136 mmol/g. In contrast to the amount of 2-NP and BPB adsorbed under the same condition (1.09 and 0.56 mmol/g, respectively), the augment in adsorption number may be due to the formation of new adsorptive sites by the adsorbed molecules. The increased total adsorbed molecule numbers indicate the feasibility of 7N-Vt in the complex binary-component system.

### 3.5. Adsorption Mechanism

#### 3.5.1. Theoretical Simulation of Molecular Orbitals and Adsorption Configuration

Electron exchange and intermolecular interactions between the adsorbent and adsorbate occur throughout adsorption processes. Investigating the electron exchange between surfactant and adsorbates, therefore, could explain the adsorption mechanism from molecular view [33]. Electron exchange is carried out in the outermost electron orbital, which can be studied by frontier molecular orbitals (FMOs). HOMO is the highest molecular orbital for providing electrons, whereas LUMO is the lowest unoccupied molecular orbital for gaining electrons [34].

Based on density functional theory (DFT), the optimized structures and FMOs distributions of surfactants and 2-NP/BPB are shown in Figure 8. The energy of FMOs is listed in Table 6. On the basis of Koopman’s theorem theory, the molecular orbital related parameters and the electron exchange number of the electron’s fraction (**Δ***N*) are calculated. The values of **Δ***N* between molecule A and B indicate the electron transfer between the two molecules: when **Δ***N* > 0, molecule A gains electrons from molecule B; when **Δ***N* < 0, molecule A donates electrons to molecule B [35].

As shown in Figure 8, the HOMO of the surfactants is concentrated in the end of the alkyl chain, and the LUMO is mainly located in the surfactant spacer, with different densities. The LUMO of 5N is covered around the whole spacer structure, distributed as two separated parts in the two headgroups of 7N, but concentrated in only one of the headgroup of 8N molecules. From the FMOs distribution of 5N, 7N and 8N, it can be seen that 7N owns the most adsorptive sites, corresponding to the highest adsorption capacity of 7N-Vt. The values of **Δ***N* between surfactants and 2-NP/BPB are all higher than 0, indicating that electrons are transferred from 2-NP/BPB to surfactant molecules. As the party for obtaining electrons, the LUMO of the surfactants accepts electrons from the HOMO of 2-NP/BPB, which acts as the main adsorptive site during the adsorption process. The energy difference between the two orbitals directly determines the energy required for electron transfer direction. The *E*_HUMO_ of 5N, 7N and 8N is −9.094, −9.016 and −8.973 eV, respectively; the *E*_LOMO_ of 2-NP and BPB is −2.006 and −1.212 eV, respectively. The electron numbers between 5N, 7N and 8N and 2-NP/BPB are 0.46/0.62, 0.39/0.53 and 0.37/0.51, respectively. Interactions between surfactants and 2-NP/BPB mainly contain hydrophobic interactions, electrostatic interactions and functional interactions (such as cationic/π and NH/π interactions). The values between **Δ***N* and 2-NP/BPB are ranked as 5N > 7N > 8N, indicating the weaker strength of functional interaction (derived from the aza group in the modifier spacer) than hydrophobic interactions. Moreover, the reverse relationship between **Δ***N* and the actual adsorption amounts may be due to 7N-Vt owning a higher modifier loading than 5N and 8N, thus providing more adsorptive sites toward 3-NP/BPB. The weakest affinity of 8N to 2-NP/BPB is demonstrated by its smallest electron exchange numbers.

The adsorption configuration and bonding length between modifiers and 2-NP/BPB are shown in Figure 8. The simulation is conducted by one 2-NP/BPB molecule reacting to one modifier molecule, to deduce the most activate site during the adsorption process. Narrower bond lengths correspond to stronger intermolecular interactions [36]. After interaction, 2-NP lie parallelly to the alkyl chain of 5N, 7N and 8N, with the nearest bond lengths being 3.90, 3.59 and 3.66 nm, respectively. In the configuration of BPB to the surfactants, however, the Br in BPB is more preferred to direct toward the N-CH_3_ group of modifiers, with the nearest bond lengths being 3.87, 3.66 and 3.69 nm, respectively. Interestingly, different locations between 2-NP and surfactant molecules may be due to the different energies of their FMOs. As shown in the left pane of Figure 8, the HOMO of the surfactants is concentrated in the end of the alkyl chain, and the LUMO is mainly located in the surfactant spacer, with different densities, while the hydroxyl group of 2-NP is part of the HOMO of the whole molecule. The LUMO of 5N, 7N and 8N is −5.29, −4.681 and −4.547 eV, respectively. It can be seen that the ΔN value between the LUMO of 5N and the HOMO of 2-NP is the smallest among three surfactants, which are lower than that between the HOMO of 5N and the LUMO of 2-NP (Table 6). Therefore, it can be speculated that interactions between the spacer in 5N and 2-NP are the weakest, which could not determine the adsorption location during the simulation process.

#### 3.5.2. Discussion of Log *K*_OM_ and Characterization of Spent Organo-Vts

The values of log *K*_OM_ for the adsorption of 2-NP/BPB on 5N-Vt, 7N-Vt and 8N-Vt are 5.12/5.04, 5.26/5.21 and 5.17/3.97, respectively (*T* = 25 °C). During the adsorption processes, higher values of log *K*_OM_ are in line with relatively stronger functional interactions [1]. Moreover, 7N-Vt shows the highest values, indicating the stronger interaction of 7N-Vt toward both 2-NP and BPB. In addition, the values of log *K*_OM_ for 2-NP adsorption are always higher than those of BPB, which is due to the more obvious effect of 2-NP diffusion and π–π stacking in the interlayer space of organo-Vts.

To further clarify the interactions during the adsorption processes of 2-NP/BPB, the FT-IR spectra of spent organo-Vts are shown in Figure 9a. Compared with organo-Vts (Figure 1c), the variation of the characteristic peaks of -CH_2_ at 2924, 2856 and 1631 cm^−1^ indicates interactions between the modifier alkyl chain and 2-NP/BPB (i.e., hydrophobic interaction and CH/π interaction). The existence of electrostatic interaction is verified by the variation of the Si-O peak from 988 to 999 cm^−1^, which demonstrates the structural effects of adsorbed 2-NP/BPB on the framework of organo-Vts [37]. The XRD patterns of the spent 7N-Vt are shown in Figure 9b. Compared with the raw 7N-Vt, the adsorption of 2-NP and BPB induces an enlargement of the interlayer spacing, demonstrating the effective diffusion of 2-NP/BPB into the interlayer space of 7N-Vt. Thereinto, the expanding effect of 2-NP on the interlayer is more obvious than that of BPB, which is due to the stronger π–π stacking effect of 2-NP [21]. In the binary-component system, the narrowest interlayer space of the spent 7N-Vt is obtained, which may be due to the competition effect of BPB and 2-NP on the diffusion of adsorbate molecules into the interlayer space of 7N-Vt, thus inducing a surface adsorption process in the binary-component system.

In conclusion, the main adsorptive sites of organo-Vts toward 2-NP and BPB are mainly on the modifier alkyl chain (hydrophobic interaction), the uncovered surface of Na-Vt (electrostatic interaction) and adsorbed adsorbate molecules (π–π interaction and π–π stacking), where 7N with a suitable interval of spacers contributes to the most adsorptive sites more than 5N and 8N. In terms of the adsorbate molecules, 2-NP with a smaller molecular size are more preferred to subject to diffusion processes into the interlayer space of organo-Vts, which is more beneficial to its higher adsorption amounts than BPB.

## 4. Conclusions

To enrich the existing organo-Vt adsorbents and to explore the effect of spacer structure on the adsorption capability of organo-Vts, a series of aza-containing gemini surfactants (5N, 7N and 8N) are applied for modifying Na-Vt, to prepare novel organo-Vt adsorbents for the first time. Successful organic modification process is verified by the structural differences between organo-Vts and Na-Vt, as unveiled by the FT-IR, XRD, TG-DTG, EA and SEM characterizations. Thereinto, 5N-Vt owns the largest interlayer spacing, the strongest binding strength, the fluffiest surface and a high modifier availability, which endow it with superiority for the adsorption of 2-NP and BPB. Excellent adsorption of 2-NP/BPB is obtained, with the maximum adsorption capacities of 5N-Vt, 7N-Vt and 8N-Vt being 142.08/364.49, 156.61/372.65 and 146.50/287.90 mg/g, respectively. All the adsorption processes are well fit by the PSO model and Freundlich isotherm and described well by the IPD model. The quicker adsorption equilibrium of 2-NP is observed than that of BPB, which is due to the easier diffusion of smaller 2-NP molecules into the interlayer space of organo-Vts. The Freundlich isotherm describes all the adsorption processes well, proving the existence of π–π stacking interaction. Electrostatic interaction plays an important role for 2-NP and BPB retention. Notably, stable regeneration of 7N-Vt is verified by the simple acid pickling method for at least three cycles, with a feasibility in the binary-component system that is demonstrated. A combination of theoretical simulation and characterization is conducted to reveal the adsorption mechanism: the adsorption processes are mainly through partition processes, electrostatic interaction and functional interactions between the modifier and the organic pollutants, where the spacer structure affects the interlayer environment and adsorptive site distribution of organo-Vts, whereas the adsorbate structure affects the diffusion process and secondary intermolecular interactions during the whole adsorption process. The results of this study demonstrate the versatile applicability of aza-based organo-Vts targeted at the removal of phenols and dyes, providing theoretical guidance for the structural optimization and mechanistic exploration of organo-Vt adsorbents.

## Figures and Tables

**Figure 1 nanomaterials-12-03636-f001:**
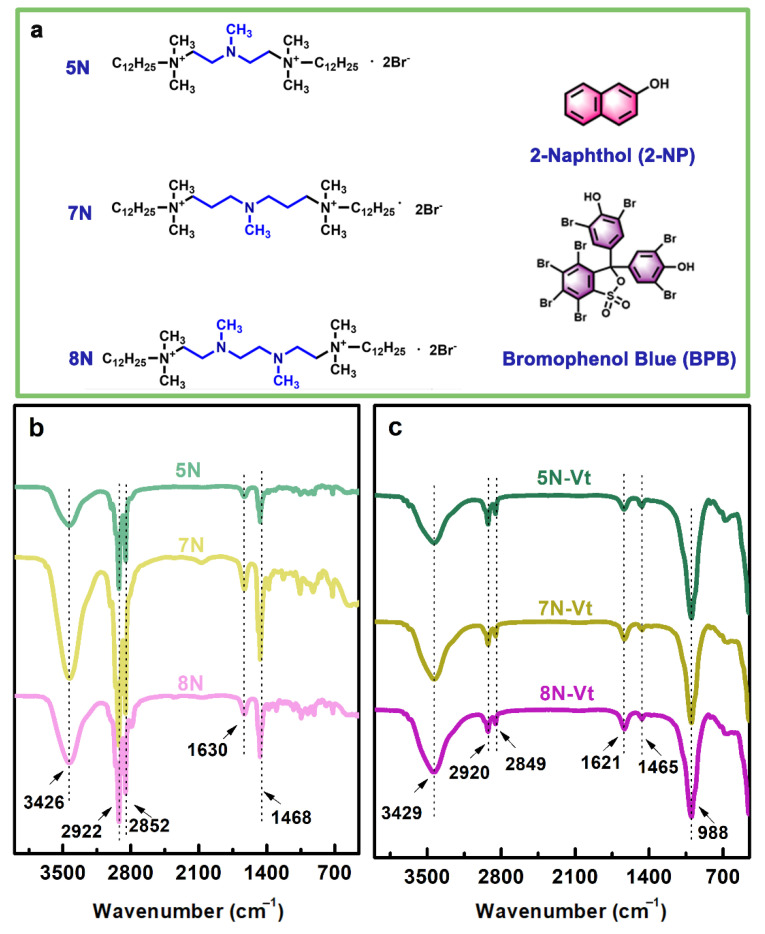
The chemical structures of surfactants, 2-NP and BPB (**a**), FT-IR spectra of surfactants (**b**) and organo-Vts (**c**).

**Figure 2 nanomaterials-12-03636-f002:**
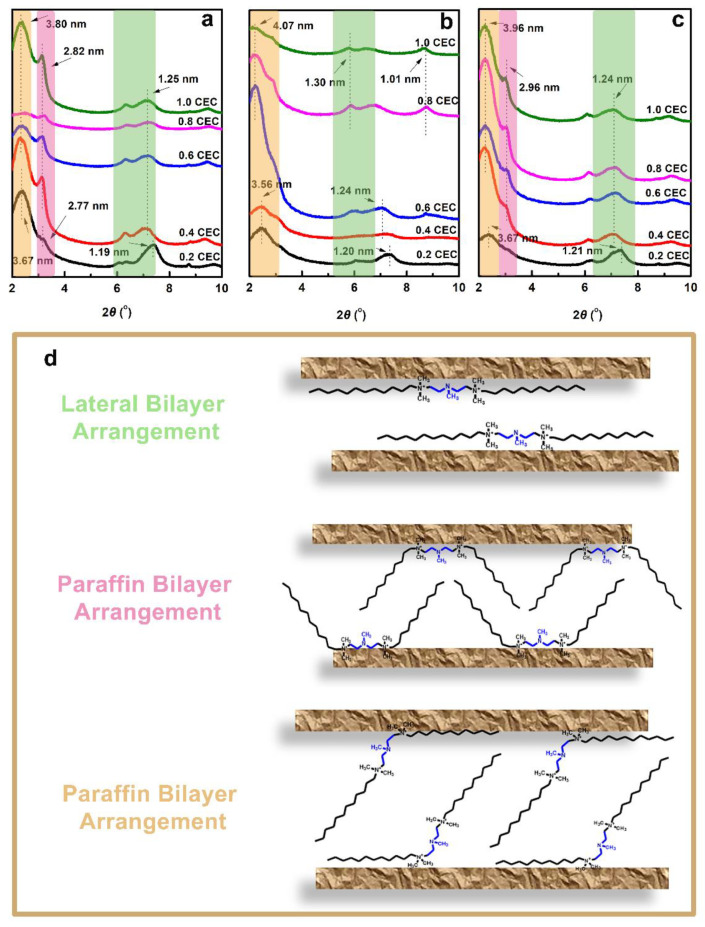
The XRD patterns of organo-Vts (**a**–**c**) and possible modifier arrangements in the interlayer space ((**d**), take 5N as an example).

**Figure 3 nanomaterials-12-03636-f003:**
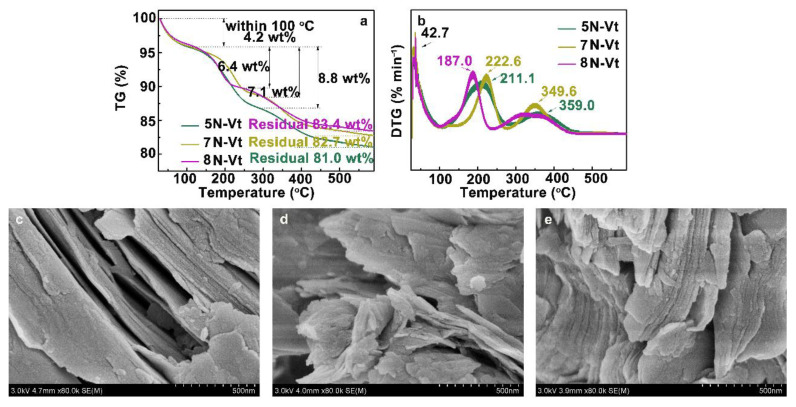
The TG (**a**) and DTG (**b**) curves of organo-Vts and SEM images of 5N-Vt (**c**), 7N-Vt (**d**) and 8N-Vt (**e**).

**Figure 4 nanomaterials-12-03636-f004:**
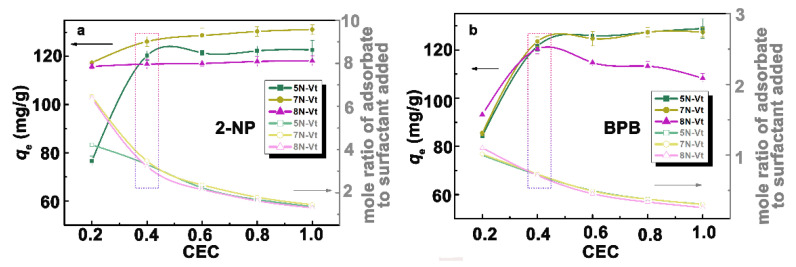
The effects of modifier dosage on the adsorption of 2-NP (**a**) and BPB (**b**) on organo-Vts (*C*_0_ = 200 mg/L).

**Figure 5 nanomaterials-12-03636-f005:**
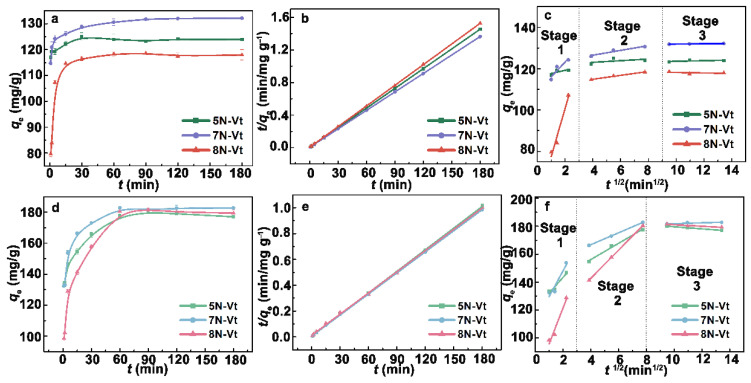
The effects of time on 2-NP and BPB adsorption (**a**,**d**), PSO kinetic models (**b**,**e**) and IPD models (**c**,**f**).

**Figure 6 nanomaterials-12-03636-f006:**
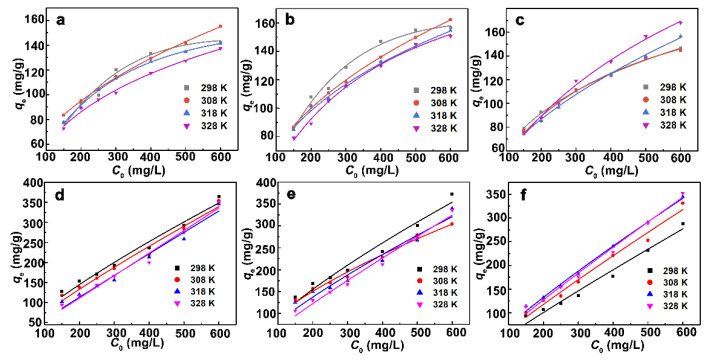
The effects of adsorbate concentration and temperature on the adsorption of 2-NP (**a**–**c**) and BPB (**d**–**f**).

**Figure 7 nanomaterials-12-03636-f007:**
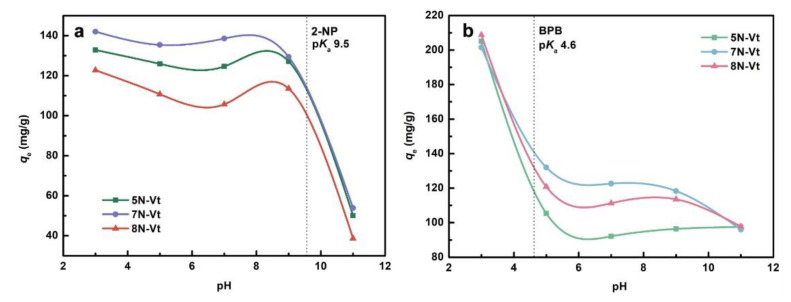
The effects of solution pH on the adsorption of 2-NP (**a**) and BPB (**b**).

**Figure 8 nanomaterials-12-03636-f008:**
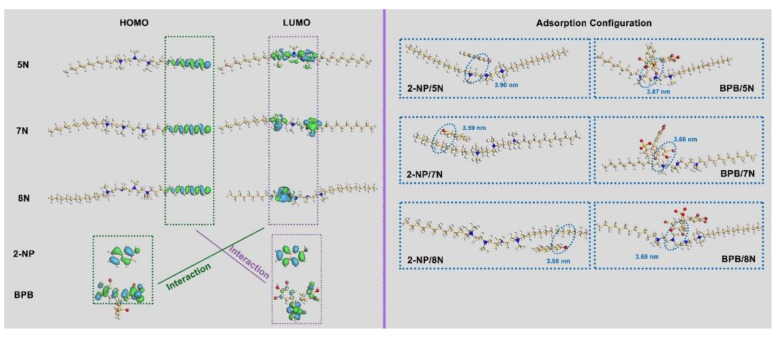
The FMOs of surfactants, 2-NP and BPB, as well as the optimized adsorption configuration of surfactants to 2-NP and BPB.

**Figure 9 nanomaterials-12-03636-f009:**
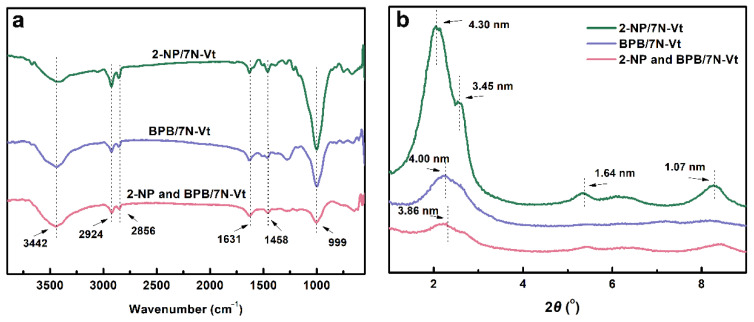
The FT-IR spectra (**a**) and XRD patterns (**b**) of adsorbed organo-Vts.

**Table 1 nanomaterials-12-03636-t001:** The elemental analysis results (wt. %) of Na-Vt and organo-Vts.

Element	Na-Vt	5N-Vt	7N-Vt	8N-Vt
**C**	0.41	12.78	12.71	12.06
**H**	0.82	2.95	2.90	2.79
**N**	0.13	1.33	1.19	1.47

**Table 2 nanomaterials-12-03636-t002:** Kinetic parameters for the adsorption of 2-NP and BPB for organo-Vts.

System	*q* _e,exp_	Pseudo-First-Order				Pseudo-Second-Order					Intraparticle Diffusion			
		*q* _e,cal_	*k*1 × 10^−2^	*R* ^2^	*SSE*	*q* _e,cal_	*k*2 × 10^−2^	*k* _2_ *q* _e_	*R* ^2^	*SSE* × 10^−5^	*k* _id_	*C* × 10^2^	*R* ^2^	*SSE*
**2NP/5N-Vt**	124.10	3.78	0.73	0.1416	5.07	123.92	9.24	11.45	0.9999	2.68	0.40	1.21	0.4122	2.8528
**2NP/7N-Vt**	132.13	9.31	2.18	0.8814	1.75	132.45	1.65	2.19	0.9999	3.18	1.16	1.22	0.8796	0.6548
**2NP/8N-Vt**	117.94	8.84	2.21	0.3886	17.80	118.34	1.83	2.17	0.9999	7.21	0.94	1.11	0.9786	0.0732
**BPB/5N-Vt**	179.95	22.74	2.41	0.3536	23.78	179.21	0.54	0.96	0.9997	26.96	5.82	1.33	0.9834	2.15
**BPB/7N-Vt**	182.73	23.78	3.10	0.6744	12.09	183.82	0.55	1.00	0.9999	6.11	4.26	1.50	0.9996	0.02
**BPB/8N-Vt**	179.39	40.04	2.50	0.5243	14.07	182.82	0.25	0.45	0.9993	59.29	10.09	1.02	0.9998	0.07

*q*_e,exp_ (mg g^−1^), *q*_e, cal_ (mg g^−1^), *k*1 (min^−1^), *k*2 (g mg^−1^ min^−1^), *k*_id_ (mg g^−1^ min^1/2^), *C* (mg g^−1^)

**Table 3 nanomaterials-12-03636-t003:** Adsorption isotherm constants for adsorption of 2-NP for organo-Vts.

Adsorbent	Temperature	Langmuir			Freundlich			Redlich Peterson			
		*q* _max_	*K* _L_	*R* ^2^	*K* _f_	*n*	*R* ^2^	*a*	*b*	*g*	*R* ^2^
**5N-Vt**	**298 K**	177.14	0.010	0.9608	20.30	3.06	0.9784	1.56	4.89 × 10^−3^	1.09	0.9534
	**308 K**	178.82	0.011	0.9438	20.24	3.01	0.9970	8.21 × 10^3^	4.05 × 10^3^	0.67	0.9962
	**318 K**	168.35	0.011	0.9931	21.07	3.18	0.9955	2.26	0.02	0.92	0.9935
	**328 K**	161.53	0.010	0.9656	17.81	3.00	0.9881	7.94	0.32	0.71	0.9861
**7N-Vt**	**298 K**	185.36	0.014	0.9758	27.97	3.44	0.9771	2.51	0.01	1.00	0.9698
	**308 K**	184.67	0.012	0.9427	22.73	3.11	0.9956	7.46 × 10^3^	3.28 × 10^2^	0.68	0.9945
	**318 K**	173.72	0.013	0.9428	24.31	3.31	0.9913	3.77 × 10^3^	1.55 × 10^2^	0.70	0.9891
	**328 K**	185.91	0.009	0.9788	17.56	2.81	0.9838	3.39	0.08	0.78	0.9856
**8N-Vt**	**298 K**	171.38	1.06 × 10^−2^	0.9677	19.61	3.04	0.9965	17.12	0.74	0.69	0.9959
	**308 K**	178.86	0.89 × 10^−2^	0.9727	16.62	2.80	0.9808	3.62	0.10	0.76	0.9801
	**318 K**	200.84	6.59 × 10^−2^	0.9605	11.72	2.36	0.9930	4.39 × 10^3^	3.74 × 10^2^	0.58	0.9913
	**328 K**	238.54	6.32 × 10^−2^	0.9790	9.70	2.12	0.9875	3.80	0.19	0.63	0.9853

*q*_max_ (mg g^−1^), *K_L_* (L mg^−1^), *K*_f_ (mg g^−1^), *a* (L g^−1^), *b* (J mol^−1^).

**Table 4 nanomaterials-12-03636-t004:** Adsorption isotherm constants for adsorption of BPB for organo-Vts.

Adsorbent	Temperature	Langmuir			Freundlich			Redlich Peterson			
		*q* _max_	*K* _L_	*R* ^2^	*K* _f_	*n*	*R* ^2^	*a*	*b*	*g*	*R* ^2^
**5N-Vt**	**298 K**	4.74 × 10^2^	0.83 × 10^−2^	0.7566	19.93	1.96	0.8660	2.02 × 10^6^	1.01 × 10^4^	0.49	0.8325
	**308 K**	6.26 × 10^2^	0.41 × 10^−2^	0.8242	10.11	1.60	0.8820	7.99 × 10^5^	7.91 × 10^3^	0.37	0.8525
	**318 K**	2.51 × 10^3^	5.42 × 10^−4^	0.8537	2.58	1.16	0.8658	4.56 × 10^3^	1.77 × 10^3^	0.14	0.8322
	**328 K**	4.14 × 10^4^	3.05 × 10^−5^	0.8851	1.68	1.06	0.8874	5.19 × 10^2^	3.09 × 10^2^	0.05	0.8592
**7N-Vt**	**298 K**	3.61 × 10^2^	0.02	0.5762	40.30	2.62	0.7882	7.83 × 10^5^	1.94 × 10^4^	0.62	0.7352
	**308 K**	3.72 × 10^2^	0.01	0.6425	29.04	2.36	0.8191	4.05 × 10^5^	1.39 × 10^4^	0.58	0.7738
	**318 K**	4.13 × 10^2^	0.86 × 10^−2^	0.6714	20.04	2.07	0.8097	8.65 × 10^6^	4.32 × 10^5^	0.52	0.7621
	**328 K**	8.78 × 10^2^	0.20 × 10^−2^	0.8852	5.33	1.38	0.9191	2.12 × 10^4^	3.98 × 10^3^	0.27	0.8988
**8N-Vt**	**298 K**	1.57 × 10^3^	6.62 × 10^−4^	0.8854	2.32	1.21	0.9047	6.06 × 10^3^	2.62 × 10^3^	0.18	0.8809
	**308 K**	9.63 × 10^2^	1.68 × 10^−3^	0.8945	4.21	1.31	0.9139	9.87 × 10^3^	2.34 × 10^3^	0.24	0.8924
	**318 K**	7.84 × 10^2^	2.87 × 10^−3^	0.9423	7.33	1.45	0.9699	2.33 × 10^4^	3.18 × 10^3^	0.31	0.9624
	**328 K**	1.12 × 10^3^	1.69 × 10^−3^	0.9066	5.02	1.32	0.9339	1.86 × 10^4^	3.70 × 10^3^	0.24	0.9174

*q*_max_ (mg g^−1^), *K_L_* (L mg^−1^), *K*_f_ (mg g^−1^), *a* (L g^−1^), *b* (J mol^−1^)

**Table 5 nanomaterials-12-03636-t005:** Thermodynamic parameters for 2-NP and BPB for organo-Vts.

System	Δ*G*° (kJ/mol)				Δ*S*° (J/(mol K))	Δ*H*° (kJ/mol)
	*T* = 298 K	*T* = 308 K	*T* = 318 K	*T* = 328 K		
**2NP/5N-Vt**	11.41	11.55	11.92	12.56	−3.76	0.10
**2NP/7N-Vt**	10.58	11.33	11.48	12.85	−6.89	0.10
**2NP/8N-Vt**	11.27	12.09	7.19	7.53	−7.71	0.60
**BPB/5N-Vt**	11.87	14.08	19.88	28.35	−5.45	−0.15
**BPB/7N-Vt**	9.69	11.79	12.57	16.95	−2.23	−0.06
**BPB/8N-Vt**	18.14	16.36	15.48	17.41	−1.34	0.06

**Table 6 nanomaterials-12-03636-t006:** Frontier orbitals’ calculation parameters of surfactants and 2-NP/BPB.

Molecules	*E*_HOMO_ (eV)	*E*_LUMO_ (eV)	Δ*N* between 2-NP	Δ*N* between BPB
**5N**	−9.094	−5.29	0.46	0.62
**7N**	−9.016	−4.681	0.39	0.53
**8N**	−8.973	−4.547	0.37	0.51
**2-NP**	−5.577	−2.006	-	-
**BPB**	−4.465	−1.212	-	-

## Data Availability

Not applicable.

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
