# Peer review of "Organo-Vermiculites Modified by Aza-Containing Gemini Surfactants: Efficient Uptake of 2-Naphthol and Bromophenol Blue"

_nanomaterials, 2022, doi:10.3390/nano12203636_

Round 1
Reviewer 1 Report
This is a nice comprehensive paper on the adsorption of organic pollutants on organo-vermiculites. Here are some remarks for improvement of the manuscript.
line 49: replace "lamellar" by "layered"; line 144: concentration measured by UV-VIS: give the wavelengths of the absorption bands and the molar extinction coefficients. Section 2.4: more details on the theoretical calculations a&re necessary; how are the structures generated? What is the size of the generated structures for the DFT calculations. line 210-211: physical interaction and electrostatic interaction: electrostatic is also physical; lines 246-247: modifier loadings: what is the CEC? What do the authors mean by modifier availability?
The adsorption process of 2NP is faster than that of BPB. Is it solely due to size differences?
Reviewer 2 Report
The manuscript titled “Organo-vermiculites modified by aza-containing gemini surfactants: Efficient uptake of 2-naphthol and bromophenol blue” uses experiment and DFT simulations to characterise the absorption of the organic pollutants, Bromophenol blue (BPB) and 2-naphthol (2-NP), onto Organo-Vermiculite (Vt) using gemini surfactants as modifiers. The authors first look at surface preparation and demonstrate that the gemini surfactants are absorbed or inserted into the surface (via surface characterisation) and suggest that these are in three possible arrangements (called lateral or paraffin). Next the authors look at the absorption of BPB and 2-NP onto the surface, considering the key environmental properties (such as time, pH and temperature), identify the saturation dosage of the modifiers and studying the differences due to dye concentration. Then, the authors investigate the reusability of the surface for repeat absorption and desorption of the dye. Finally, there is a theoretical study using DFT to understand the absorption mechanisms.
Overall, I found the article and supplemental information well written, with a good level of detail using appropriate characterisation methods and is suitable for publication in the Nanomaterials Special issue. The authors appear to have considered all key aspects of the process and it is interesting to see that they have included a study of reusability which would be important for the practical application.
My comments on the text are as follows:
Line 150: The volume units of V or C seem inconsistent for the Eq. 1, shouldn’t they either both be L or mL?
Line 452: I believe this line may have two typos and should read:
“The ELUMO of 5N, 7N and 8N are -9.094,”…
Table 3&4: Some column entries contain bracket terms whose meaning is unclear to me: what do these mean are the authors saying multiply the answer by the given power of 10 or something else?
Table 5: Column heading for the temperature seems to be misaligned.
Line 417: I not sure I understand the meaning “is suffered a lot”?
In Figure 8 (right pane) the NP molecule is shown at very different locations with respect to the spacer for 5N ,7N and 8N. Can the authors explain why as I’d expect these locations to be similar? Specifically, the hydroxyl group of NP is pointing towards the central spacer units of 5N but away in 7N and 8N.
Reviewer 3 Report
The study is focused on the possible applicability of aza-based organo-Vts in view of removal of phenols and dyes. Options for structural optimization are discussed aimed at investigating the effects of spacer structure on the adsorption capability of organo-Vts through application of a series of aza-containing gemini surfactants (5N, 7N and 8N) for the modification of Na-Vt.
The obtained results are of interest to the readers of MDPI Nanomaterials, but there are some issues which have to be resolved before accepting the manuscript:
1. There are ambiguous statements and unclear/wrong terminology, e.g.:
1.1 line 230: “The combination force between surfactants and organo-Vts contains physical interaction and electrostatic interaction….” The terms ‘combination force’ and ‘physical interaction’ are unclear and dubious.
1.2. lines 237-239: “…the destroy of electrostatic interactions between modifiers and the surface of Na-Vt, whose strength is affected by the steric hindrance of the head group within surfactant…”
1.3. lines 289-290: “…the modifier dosage of 0.4 CEC is chosen as the saturated amount…” What does ‘saturated amount’ mean?
1.4. line 358: “… the secondary interaction between BPB molecules” What does ‘secondary interaction’ refer to?
1.5. line 416: “…while that of BPB is suffered a lot…”
2. The texts in Figures 3a, 3b, 4, 5 and 6 are too small and unreadable.
3. Short descriptions of the physical backgrounds for “pseudo-first order (PFO), pseudo-second-order (PSO) and intra particle diffusion (IPD) models” (lines 296-298) should be added.
Round 2
Reviewer 3 Report
Ok, accept in present revised form.